# Adults of Alderflies, Fishflies, and Dobsonflies (Megaloptera) Expel Meconial Fluid When Disturbed

**DOI:** 10.3390/insects14010086

**Published:** 2023-01-13

**Authors:** Pei Yu, Chengquan Cao, Xingyue Liu, Fumio Hayashi

**Affiliations:** 1Department of Biology, Tokyo Metropolitan University, Minamiosawa 1-1, Hachioji 192-0397, Tokyo, Japan; 2College of Life Science, Leshan Normal University, Leshan 614004, China; 3Department of Entomology, China Agricultural University, Beijing 100193, China

**Keywords:** anti-predatory behavior, defensive secretion, defensive spray, meconium, potential predators

## Abstract

**Simple Summary:**

In most holometabolous insects, meconial fluid produced via the Malpighian tubules at the pupal stage is all discharged from the anus at adult emergence. However, adult alderflies (Sialidae), fishflies (Corydalidae: Chauliodinae), and dobsonflies (Corydalidae: Corydalinae) of the Megaloptera order are likely to expel the meconial fluid when they are disturbed after emergence. In this study, we compare this fluid expelling behavior among 1 species of the alderfly, 12 species of four fishfly genera, and 15 species of three dobsonfly genera. The response of expelling to artificial stimulation occurs less frequently and the amount of expelled fluid is relatively smaller in alderflies than in the other groups. *Parachauliodes* fishflies and *Protohermes* dobsonflies expel relatively larger amounts of fluid than the other genera of Corydalidae observed herein. The vigorous expelling of fluid when they are disturbed may have the function of predation avoidance. In the future, we need to examine how this behavior is effective at deterring predators and why it develops more in some groups of Megaloptera.

**Abstract:**

Chemical secretions are an effective means by which insects may deter potential enemies, already being studied extensively with regard to their chemicals, synthesis, toxicity, and functions. However, these defensive secretions have been little studied in Megaloptera. Herein, the fluid expelling behavior of adult alderflies (Sialidae), fishflies (Corydalidae: Chauliodinae), and dobsonflies (Corydalidae: Corydalinae), all of the order Megaloptera, is described in detail regarding the timing and possible function of this behavior. When disturbed artificially, both males and females could expel fluid from the anus. However, the frequency of expelling was much lower in alderflies than in fishflies and dobsonflies. The amount of expelled fluid relative to body weight was also smaller in alderflies. In fishflies and dobsonflies, the amount of expelled fluid decreased with adult age, probably because the fluid is little replenished once expelled. The cream-colored fluid seems to be meconial fluid produced via the Malpighian tubules at the pupal stage, which is usually discharged at adult emergence in most other holometabolous insects. However, adult fishflies and dobsonflies often expel it vigorously by bending their abdomen when disturbed after emergence. Thus, the fluid expelling may be an anti-predatory behavior, particularly in younger adults that can expel a relatively large amount of fluid.

## 1. Introduction

Insects are physically and/or chemically protected against predation [1,2,3,4,5,6,7]. Insects should invariably attempt to avoid capture by predators. Escape, however, is not always possible. Once seized by the predator, the prey has a variety of options available [8]. Expelling meconial fluid might be one example of an escaping behavior, because it occurs when adults are disturbed by physical stimulation, such as the grasping of their body [8,9]. Metabolic wastes accumulated during pupal development are collected via the Malpighian tubules and are emptied into the midgut with the filtrate of fluid and ions as meconium [10,11,12]. In Lepidoptera and some Neuroptera species, a considerable amount of the meconial fluid is secreted from the anus from just-emerged adults [8,13,14,15]. Moths and butterflies can expel this fluid against predators during the short, vulnerable post-ecdysial period [8], but adult moths of *Euxoa auxiliaris* (Grote) can expel it in response to a disturbance after emergence, although the expelled fluid peaks in quantity just after the emergence and decreases with age [9]. The adult moth *Arctia plantaginis* (Linnaeus) also expels the majority of their anal fluid in the first-day attack and lowers quantities during the second- and third-day attacks [13].

In Megaloptera, meconial fluid is also expelled from the anus at emergence [16], and *Neohermes californicus* (Walker) is described to void white meconium by just-emerged adults and field-captured adults when handled [17]. We found this expelling behavior of meconial fluid in a wide range of adult alderflies (Sialidae), fishflies (Corydalidae: Chauliodinae), and dobsonflies (Corydalidae: Corydalinae), all members of Megaloptera, when they are disturbed with hands in the laboratory (Figure 1). Megaloptera consists of the superorder Neuropterida, together with the orders of Raphidioptera and Neuroptera. Neuroptera insects are also known to produce meconium at adult eclosion, but the properties of it differ among the taxonomic groups [14,15,18]. Most families of Neuroptera deposit a pellet of meconium (not fluid), but the families Berothidae and Mantispidae expel brownish liquid meconium at eclosion. In this study, first, the meconial fluid expelling behaviors of adult alderflies, fishflies, and dobsonflies are described, and second, the frequency and the amount of expelled fluid are compared on their molecular phylogenetic tree. Finally, the effect of adult ages on this behavior is examined. The function of meconial expelling as a defense against predators is discussed.

## 2. Materials and Methods

### 2.1. Fluid Expelling Behavior

The behavior of expelling fluid was observed by stimulating adults with human hands for 1 species of alderfly, 12 species of four fishfly genera, and 15 species of four dobsonfly genera (Table 1). Adults were obtained by light traps and a sweeping net in the field and rearing the field caught last-instar larvae, prepupae, and pupae. The larvae caught in winter were placed individually in glass vessels (70 mm in diameter and 90 mm in height) with small stones as refuges on the bottom. Well-aerated tap water, not exceeding 5 mm in depth, was supplied and replaced daily. Larvae were given one or two living last (4th)-instar chironomid larvae every day. The rearing vessels were kept in an incubator at a constant temperature of 15 ± 1 °C (13 h light and 11 h dark light cycle) until April and 20 ± 1 °C thereafter (14 h light and 10 h dark light cycle). Approximately a week after the larvae stopped feeding, they were individually relocated into same-sized vessels containing moist peat moss, in which they made a space for pupation. These vessels were kept at 25 ± 1 °C (14 h light and 10 h dark light cycle). The field-caught prepupae and pupae were also maintained individually in such peat moss.

In general, adults emerged after about the 10-day prepupal and 10-day pupal period. Newly emerged adults were reared individually in glass vessels (70 mm in diameter and 90 mm in height) at 25 ± 1 °C (14 h light and 10 h dark light cycle). The glass vessels contained wet filter paper on the bottom to prevent desiccation and nylon mesh covering the top. Adults were given a sufficient amount of a 10% sucrose solution every day.

Fluid expelled from the anus was sampled using a known-weight plastic bag, in which the adult thoracic part was grasped directly, with hands at the mouth of the bag. To avoid different responses depending on the handling pressure, only the first author grasped the adult thoracic parts throughout this study. Immediately after expelling, the adult was removed, and the bag was weighed again. Then, the amount of expelled fluid was calculated as the weight difference of the bag before and after expelling. This experiment was performed before giving the sucrose solution on the day, and once used, individuals were not examined again. The adult head width (between the outer margins of the right and left eyes) was measured using a digital caliper. To remove the effect of body size, the residuals of the expelled fluid weight were plotted on their phylogenetic tree [19].

### 2.2. Age-Dependent Fluid Expelling

The effect of adult age after eclosion on the amount of expelled fluid was examined from 2017 to 2021 for four species of fishflies, namely *Parachauliodes continentalis* (from Miyagi, Japan), *P. asahinai* (from Fukui, Japan), *Neochauliodes amamioshimanus* (from Amamioshima Island, southern Japan), and *N. occidentalis* (from Sichuan, China), and seven species of dobsonflies, namely *Protohermes davidi* (from Sichuan, China), *P*. *grandis* (from Tokyo, Japan), *P. immaculatus* (from Amamioshima Island, southern Japan), *P. guangxiensis* (from Sichuan, China), *P. similis* (from Sichuan, China), *P. horni* (from Sichuan, China), and *Neoneuromus maclachlani* (from Sichuan, China). Fluid expelled from the anus was sampled using a known-weight plastic bag, in which the adult thoracic part was grasped with hands and weighed. This experiment was performed once for laboratory bred individual adults of different ages.

### 2.3. Replenishment of Expelled Fluid

The recovery rate of fluid to expel was examined from 2018 to 2021 for three species of fishflies, namely *Parachauliodes continentalis* (from Miyagi, Japan), *P. japonicus* (from Saitama, Japan), and *Neochauliodes amamioshimanus* (from Amamioshima Island, southern Japan), and seven species of dobsonflies, namely *Protohermes davidi* (from Sichuan, China), *P*. *grandis* (from Tokyo, Japan), *P. immaculatus* (from Amamioshima Island, southern Japan), *P. similis* (from Sichuan, China), *P. weelei* (from Yunnan, China), *Neoneuromus indistinctus* (from Yunnan, China), and *N. maclachlani* (from Sichuan, China). The adult thoracic region was grasped by hands, and the amount of expelled fluid was weighed on the day of emergence (day 0), on the 5th day, and on the 10th day for the same individuals.

### 2.4. Statistics

In a log–log scatterplot between individual body size (head width) and fresh weight of expelled fluid, Pearson’s correlation analysis was used, and then, the residual of the log-scale fluid weight was calculated from the regression equation for comparison among the taxa. The Kruskal–Wallis test and Steel–Dwass multiple comparison test were used for the comparisons of the residuals among the three groups of taxa, and the Mann–Whitney *U*-test was used between the two groups of taxa. Spearman’s rank correlation coefficient tests were used to examine the relationship between the adult age (days after emergence) and the amount of expelled fluid in each species. The relationship between adults’ age and adults’ body size (head width) was also tested using Spearman’s rank correlation coefficient in each species, to check whether the effect of adults’ body size on the amount of expelled fluid should be excluded or not. The amount of expelled fluid was compared among the groups of adults aged 0, 5, and 10 days using the Friedman test, followed by the Dunn–Bonferroni post-hoc test to determine which group combinations had a significant difference. These tests were performed using IBM SPSS Statistics ver. 2.5.

## 3. Results

We observed that most adults expelled fluid from the anus when their body was grasped by hands (Figure 1; also see the free video in Movie Archives of Animal Behavior: http://www.momo-p.com/showdetail-e.php?movieid=momo211129mn01b&embed=on). The frequency of expulsion was much lower in alderflies than in fishflies and dobsonflies (Table 1). They expelled fluid toward the hands by bending their abdomen if abdominal movement was not limited physically by the hands. The fluid was usually cream-colored but was occasionally transparent, whitish, or slightly reddish, varying even in the same species.

The body size (head width) varied among the species and genera (Table 1). The alderflies were smaller than the fishflies and dobsonflies. The amount of expelled fluid also varied among the species and genera, as well as among conspecific individuals (Table 1). The fresh weight of fluid (*y* mg) expelled by individuals positively correlated with their head width (*x* mm) in the log–log relationship (log_10_ *y* = 3.159 log_10_ *x* − 1.614, *r* = 0.514, *df* = 277, *p* < 0.001) in all combined data (*n* = 279, excluding 2 *Parachauliodes japonicus*, 3 *P. continentalis*, 1 *Neochauliodes occidentalis*, 1 *N. amamioshimanus*, 1 *Protohermes davidi*, 2 *P. grandis*, and 1 *Neoneuromus maclachlani* lacking data on head width among the 290 expelling individuals in Table 1). The residuals of the expelled fluid weight calculated from this equation differed between species and genera (Figure 2). In fishflies, the expelled amounts of five species of the closely related *Nigronia* and *Parachauliodes* were larger than those of six species of *Neochauliodes*, but the difference was not significant statistically (Mann–Whitney *U*-test; *z* = 1.83, *n*_1_ = 5, *n*_2_ = 6, *p* = 0.068). This amount differed among the three dobsonfly genera *Protohermes* (nine species), *Neoneuromus* (four species), and *Acanthacorydalis* (two species) (Kruskal–Wallis test; *χ*^2^ = 7.41, *df* = 2, *p* = 0.025), and *Protohermes* expelled relatively more fluid than *Neoneuromus*, although it was statistically marginal (Steel–Dwass multiple comparison test; *p* = 0.050).

The amount of fluid expelled when stimulated by hands decreased as the age of the adult advanced, although in some species, particularly those with a small sample size, this tendency was not statistically significant (Figure 3). In this experiment, there was no significant correlation between the age of the adult and the head width, except for the male *Neochauliodes occindentalis* (*r*_s_ = −0.73, *df* = 8, *p* = 0.02), in which the decline in expelled fluid with age may be a side effect of their body size.

Repeated stimulation by hands to the same individuals at 0, 5, and 10 days after emergence resulted in a great decrease in the expelled fluid after the first stimulation, although this tendency was not statistically significant in some species (Figure 4). The expulsion fluid seemed to be little replenished.

## 4. Discussion

The expelling of meconial fluid from the anus by adults has been reported in several holometabolous insects [8,9,13,14,15], including the first report of it in the Megaloptera fishfly *Neohermes californicus* [17]. In this study, we confirmed such a behavior in a wide range of Megaloptera and revealed some species- or genus-specific differences in its frequency and quantity. The frequency of fluid expelling when disturbed, and the amount of expelled fluid relative to the body size were much lower in alderflies than in fishflies and dobsonflies (Table 1, Figure 2). Alderflies are the smallest in body size among the Megaloptera. In contrast, fishflies and dobsonflies, which are much larger than alderflies, expelled fluid vigorously when their thorax region was grasped with hands (Figure 1). They can expel it in various directions by bending their abdomens, usually toward the stimulus (hands in our observations). In contrast to the secretion from the anus by Megaloptera, several insects discharge fluids from specialized glands when disturbed. This has been shown for earwigs of *Doru* that have a pair of defensive glands on the fourth abdominal tergite from which they discharge fluid when disturbed [20] and cockroaches of the genera *Eurycotis*, *Deropeltis*, and *Diploptera* that have glands that open as slits on their abdominal segments and spray fluid [4]. These insects can control the direction toward which the fluid is emitted using their abdomens [4,20].

The fluid expelled by alderflies, fishflies, and dobsonflies is similar in color to the meconium expelled by the moths *E. auxiliaris* and *A. plantaginis* when disturbed [9,13]. In alderflies, fishflies, and dobsonflies, a small amount of cream-colored fluid was observed occasionally on the substrates and around the pupal exuviae in the rearing vessels at the adult emergence. This was also similar in color to the expelled fluid and the Malpighian tubule contents of adults when dissected. These facts suggest that the fluid expelled by disturbed adults consists primarily of meconial fluid, although further chemical analysis is needed to confirm whether other chemical compounds are included or not.

In fishflies and dobsonflies, the expelled fluid peaked in quantity on the emergence day and decreased with age (Figure 3 and Figure 4). Once this fluid was expelled, it was little replenished. Similar results were reported in the meconial expelling moth *E. auxiliaris* [9]. After ejecting once in response to a disturbance, this moth cannot immediately expel again, and has slightly lower capacities for discharge at 16, 58, and 82 h after the first ejection. In the moth *A. plantaginis*, the majority of anal secretion was released in the first-day attack, with much lower quantities during the second- and third-day attacks [13]. Thus, the fluid expelled by these moths seems to mostly occur once when young, as in the case of fishflies and dobsonflies, possibly to escape predators.

Fishflies and dobsonflies are relatively large insects (Figure 1, Table 1). They rest on tree trunks, branches, and leaves during the daytime but fly actively at night [21,22]. Their main potential predators may be birds and other vertebrates approaching in the daytime, and flying bats and nocturnal birds such as owls and nightjars at night. In fact, the megalopterans are listed in their diets, as revealed by a recent molecular analysis of fecal samples of bats [23,24,25], nocturnal birds [26], and diurnal birds [27]. A vigorous expulsion of fluid seems to affect predators to hesitate to attack the prey. If the predators hesitate, the prey could fly away without severe injuries. Thus, fluid expelling itself may be a deterrent against potential predators. Moreover, enriched fluid expelled to birds and bats may sometimes facilitate escape if it reaches the predator’s eyes [2]. At night, bats detect flying insects using echolocation systems and usually scoop them with their wings or other parts of their bodies to bite them [28]. In this case, expelling fluid at the time of scooping is expected to be efficient in making bats hesitate to bite the prey. Such a situation may occur when insectivorous lizards, birds, and monkeys attack insect bodies and wings with their mouth, beak, or hands in the daytime. The grasping of the thoracic region in our experiments may be similar to the stimulus by the predators of Megaloptera.

Vigorous fluid expulsion itself may have the effect of discouraging predators. The use of stored metabolic wastes (meconial fluid) may be a costless method. Adult fishflies and dobsonflies have relatively large and hard mandibles among insects [29], which allow them to inflict strong bites on predators (and can draw blood from human handlers). Thus, these indirect (vigorous expelling) and direct (biting) defensive behaviors may increase survival against various insectivorous predators. We need further information on the morphology, physiology, and behavior in genera and species of Megaloptera to understand if there are avoidance mechanisms against predators.

## 5. Conclusions

Adults of the Megaloptera family are capable of expelling fluid vigorously from the anus when disturbed with hands as a simulation of attack by potential predators. This fluid is cream-colored and is considered to be meconium produced via the Malpighian tubules at the pupal stage. This fluid is usually discharged at adult emergence in most holometabolous insects, but the megalopteran adults seem to use it as a defensive fluid. Adult alderflies and some fishfly and dobsonfly genera expel a small amount of fluid relative to their body size. The amount of the expelled fluid also decreases with adult age after emergence, probably because the fluid is little replenished once expelled. The effectiveness of this megalopteran fluid expelling behavior has still not been examined, but this behavior may function as an anti-predatory mechanism. More information on the chemicals present in the expelled fluid and direct evidence of the effects against potential predators are needed.

## Figures and Tables

**Figure 1 insects-14-00086-f001:**
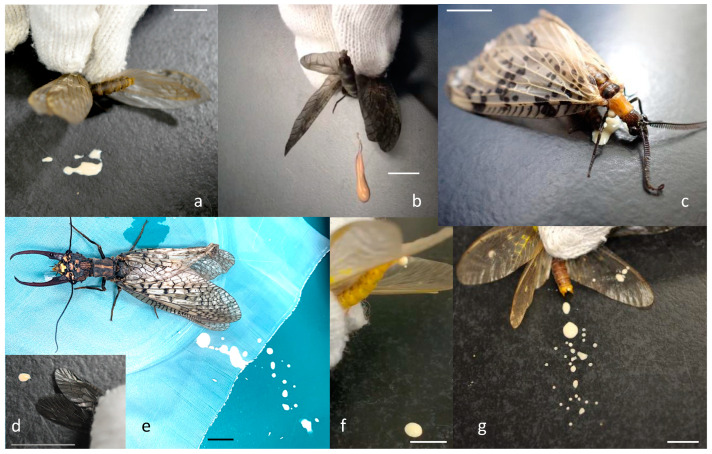
The fluid expelling behavior just after grasping by the thorax region of individual alderflies ((**d**): *Sialis melania*), fishflies ((**a**): *Parachauliodes continentalis*, (**b**): *P. japonicus*, and (**c**): *Neochauliodes amamioshimanus*), and dobsonflies ((**e**): *Acanthacorydalis orientalis*, (**f**): *Protohermes immaculatus*, and (**g**): *P. grandis*). Scale bars: 10 mm. For movies of this behavior, click http://www.momo-p.com/showdetail-e.php?movieid=momo211129mn01b&embed=on.

**Figure 2 insects-14-00086-f002:**
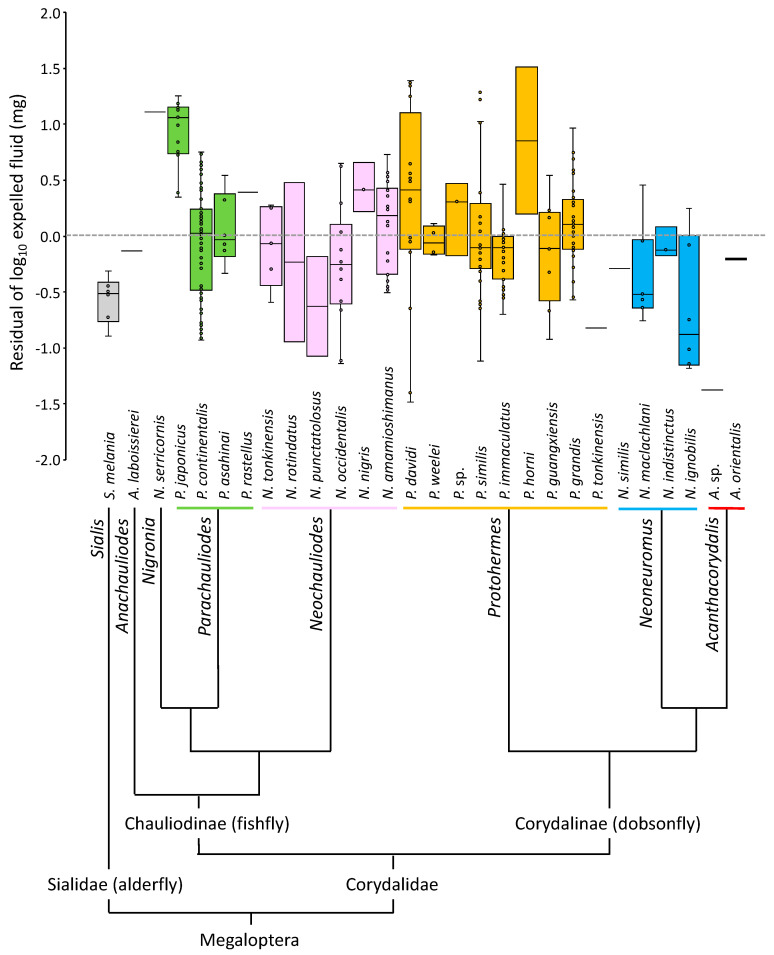
The box plots of the relative weight of the expelled fluid to body size in 1 species of alderflies, 12 species of fishflies and 15 species of dobsonflies plotted on their phylogenetic tree at the genus level [19]. For an explanation of the log-scale residual of the expelled fluid weight, see the statistics section in the main text.

**Figure 3 insects-14-00086-f003:**
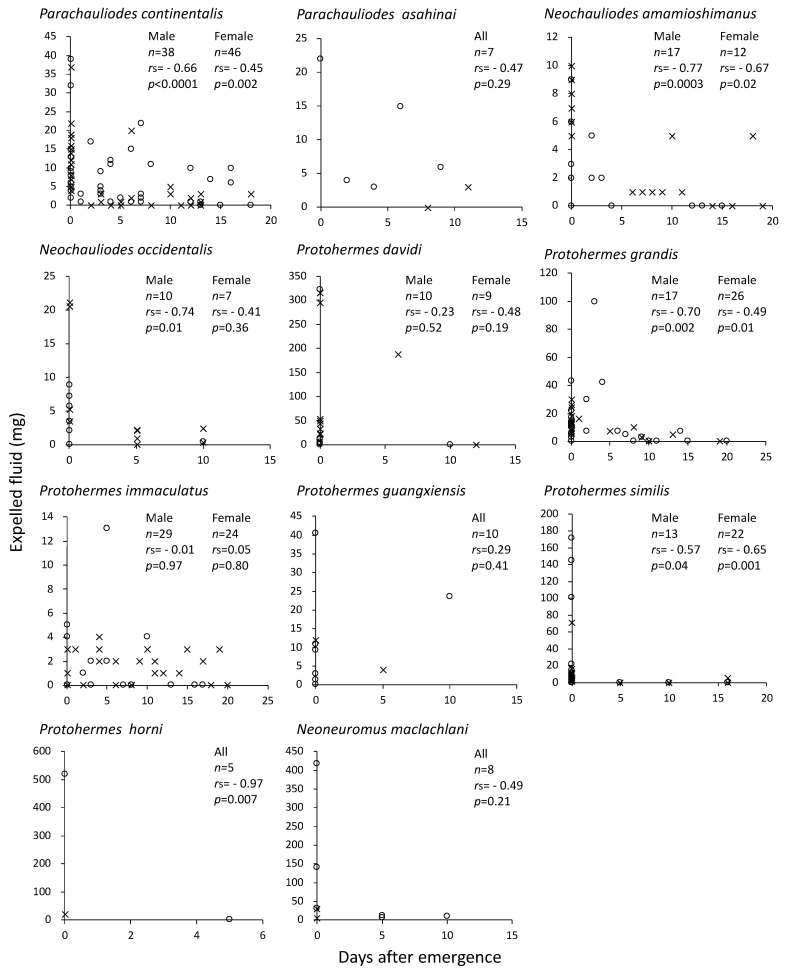
The relationships between adult ages and the amount of expelled fluid in 11 species of male (x) and female (o) fishflies (*Parachauliodes*, *Neochauliodes*) and dobsonflies (*Protohermes*, *Neoneuromus*). *n*: number of individuals examined. *r*_s_: Spearman’s rank correlation coefficient. Male and female data are combined in the species with small sample size.

**Figure 4 insects-14-00086-f004:**
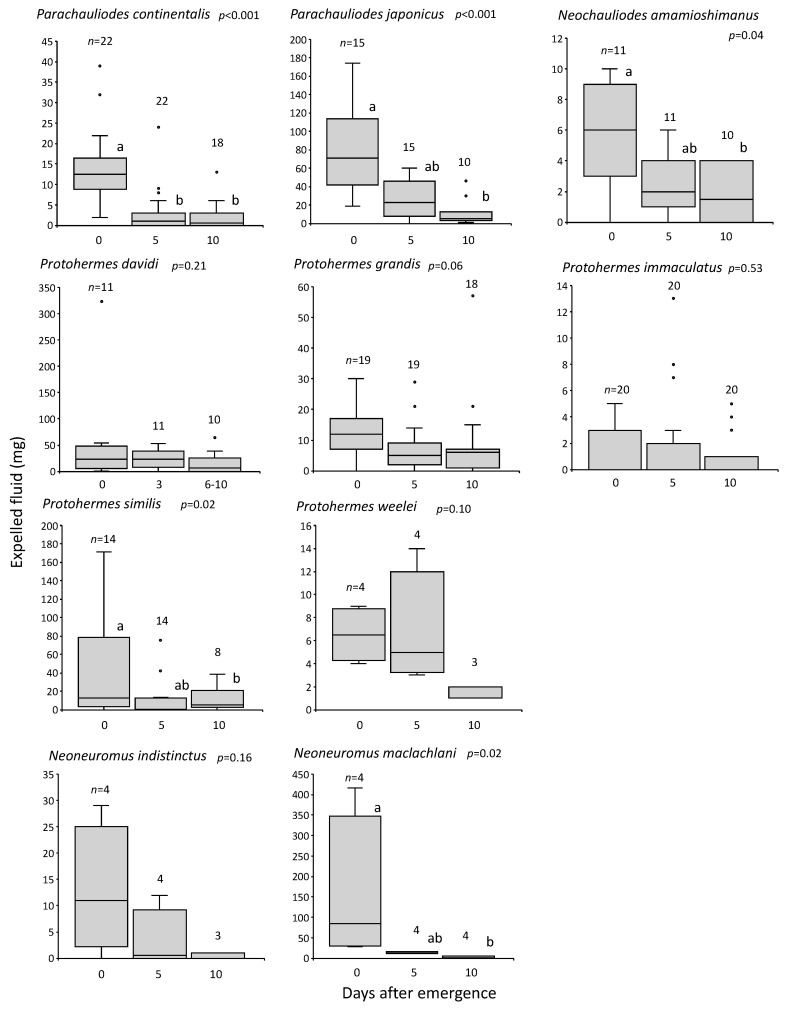
The amount of fluid expelled at 0, 5, and 10 days from emergence (but 0, 3, and 6–10 days in *P. davidi*) in 10 species of fishflies (*Parachauliodes*, *Neochauliodes*) and dobsonflies (*Protohermes*, *Neoneuromus*). *n*: number of individuals examined, but some individuals died at the last examinations. Crosses are the mean values. *p*: the result of the Friedman test. Different alphabetic characters show significant differences (*p* < 0.05) in the Dunn–Bonferroni post-hoc test in the species with statistically significant differences in the former test.

**Table 1 insects-14-00086-t001:** Information of localities, adult stages, male and female size (head width), and fluid expelling behaviors in each species of Megaloptera examined in this study. *n*: number of individuals examined. SE: standard errors, but the range at *n* = 2. For more information of the amount of fluid expelled by individuals among taxa, see Figure 2.

Family	Species	Locality	Adult Stage	Male Head Width (mm)			Female Head Width (mm)			Fluid-Expelling Behavior		
(Subfamily)				*n*	Mean	SE	*n*	Mean	SE	*n*	Expelled Individuals	Range of
											(%)	Expelled Fluid (mg)
Sialidae	*Sialis melania* Nakahara	Japan	field collected	8	2.26	0.04	26	2.77	0.03	34	6 (17.6)	0.1–0.2
Corydalidae	*Anachauliodes laboissierei* (Navás)	China	reared	1	5.51		0	–		1	1 (100)	3.9
(Chauliodinae)	*Nigronia serricornis* (Say)	N. America	reared	0	–		1	4.00		1	1 (100)	25.0
	*Parachauliodes japonicus* (McLachlan)	Japan	reared	3	5.73	0.19	10	6.30	0.12	15	15 (100)	19.0–174.0
	*Parachauliodes continentalis* van der Weele	Japan	reared	32	5.53	0.04	46	6.14	0.04	83	69 (83.1)	1.0–39.0
	*Parachauliodes asahinai* Liu, Hayashi & Yang	Japan	reared	2	5.18	0.13	5	5.74	0.12	7	6 (85.7)	3.0–22.0
	*Parachauliodes rastellus* Shimonoya	China	field collected	0	–		1	6.10		1	1 (100)	18.0
	*Neochauliodes tonkinensis* van der Weele	China	field collected	6	5.20	0.11	0	–		6	5 (83.3)	1.3–9.1
	*Neochauliodes rotundatus* Tjeder	China	reared	3	4.50	0.11	0	–		3	2 (66.7)	0.3–7.7
	*Neochauliodes punctatolosus* Liu & Yang	China	field collected	0	–		2	4.84	0.01	2	2 (100)	0.3–2.3
	*Neochauliodes occidentalis* van der Weele	China	reared	10	5.31	0.08	6	5.48	0.16	17	15 (88.2)	0.3–21.2
	*Neochauliodes nigris* Liu & yang	China	field collected	5	4.86	0.12	0	–		5	3 (60.0)	7.2–17.0
	*Neochauliodes amamioshimanus* Liu, Hayashi & Yang	Japan	reared	16	4.33	0.05	12	4.64	0.08	29	21 (72.4)	1.0–10.0
(Corydalinae)	*Protohermes davidi* van der Weele	China	reared	10	7.07	0.11	8	7.45	0.10	19	17 (89.5)	0.4–323.4
	*Protohermes weelei* Navás	China	reared	0	–		4	5.96	0.09	4	4 (100)	4.0–9.0
	*Protohermes* sp.	China	reared	1	5.20		2	5.90	0.60	3	3 (100)	6.0–14.0
	*Protohermes similis* Yang & Yang	China	reared	13	6.03	0.03	22	6.50	0.05	35	25 (71.4)	0.7–171.7
	*Protohermes immaculatus* Kuwayama	Japan	reared	29	4.68	0.03	24	5.27	0.04	53	27 (50.9)	1.0–13.0
	*Protohermes horni* Navás	China	reared	1	7.23		4	7.97	0.30	5	2 (40.0)	19.7–518.6
	*Protohermes guangxiensis* Yang & Yang	China	reared	2	6.33	0.04	8	7.27	0.10	10	8 (80.0)	1.3–40.4
	*Protohermes grandis* (Thunberg)	Japan	reared	15	6.17	0.06	24	6.80	0.07	43	35 (81.4)	3.0–100.0
	*Protohermes tonkinensis* (van der Weele)	China	field collected	2	4.43	0.40	0	–		2	1 (50.0)	0.3
	*Neoneuromus similis* Liu, Hayashi & Yang	China	field collected	4	9.67	0.07	0	–		4	1 (25.0)	17.0
	*Neoneuromus maclachlani* (van der Weele)	China	reared	2	8.93	0.79	5	10.22	0.24	8	8 (100)	4.2–416.0
	*Neoneuromus indistinctus* Liu, Hayashi & Yang	China	reared	2	7.70	0.30	2	8.65	0.20	4	3 (75.0)	9.0–29.0
	*Neoneuromus ignobilis* Navás	China	field collected	0	–		8	9.86	0.24	8	6 (75.0)	2.2–48.9
	*Acanthacorydalis* sp.	China	field collected	0	–		1	12.26		1	1 (100)	2.8
	*Acanthacorydalis orientalis* (McLachlan)	China	reared	2	15.18	0.77	0	–		2	2 (100)	68.5–97.5

## Data Availability

The data presented in this study are available on request from the authors.

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
