# Peer review of "Adults of Alderflies, Fishflies, and Dobsonflies (Megaloptera) Expel Meconial Fluid When Disturbed"

_insects, 2023, doi:10.3390/insects14010086_

Round 1

Reviewer 1 Report (Previous Reviewer 1)

Congratulations on the revised version of the manuscript, and about the study itself, quite original and opens several future lines of research.  My only recommendation is to apply a last and quick English style checking by a native speaker, however writing is overall acceptable.

Author Response

Dear reviewer, we are deeply grateful for your timely and careful revision.

We have revised all points based on the comments written in the file by you.

Reviewer 2 Report (Previous Reviewer 2)

Adults of Alderflies, Fishflies and Dobsonflies (Megaloptera) Expel Fluid When Disturbed

General comments on a resubmission:

The authors modified some parts of the manuscript according to the reviewers comments on the first version (others not). However, the general problem in the paper is not solved. Even terms have been changed, the approach described is still based on the idea of chemical defence by meconial fluid. The rationale is confusing because expellation of meconial fluid is compared to defecation / regurgitation after feeding as well as excretion from specialized glands.

I suggest to mention 'meconial fluid' in the title  to clearly indicate the content of the paper. The text should avoid the idea  of (chemical) defense except as a hypothesis in discussion.

Specific comments:

see also my comments on first version.

Line 26 ff: It is obvious that the amount of expelled fluid is less in small species.

Line 30 ff: no data is given about storage of fluid and expelling when stimulated by predators. This should be clearly described as authors hypothsis.

Line 57: meconial secretion of Corydalidae after disturbance was reported by Smith, 1970. However, why do the authors expect this is an exception.

Line 101: I do not understand the technique well. Does it mean the specimen was put into a plastic bag and thorax was grasped /squeezed from out site the bag?

Table 1

SEa – I don’t see the explanation of the footnote.

n in the three columns is still not understandable. In line 104 authors say that the head width of each adult was measured. This does not fit to tab. 1.

For example P. japonicus:

Head width of 3 males + 10 females were measured but 15 individuals were used in the expelling experiment and each individual expelled successfully. Which individuals were excluded from the measurement and why? If head width of some individuals is missing, this should be mentioned .

Line 145: see my comments on Tab 1.

Was the correlation calculated across all species? This is supported by df = 277 in line 170. However Tab. 1 shows 287 expelled individuals.

Line 158: Figure 1 does not show the described technique in Material and Methods.

Line 166 ff: see my comments on Tab 1. Which individuals were excluded from the correlation?

Line 172: the term … seemed to be greater … is not aplicable for a statistical analysis. The result should be described correctly.

Line 177: what is the rational to test the correlation between adult age and heads width. The negative rs for males of one species has to be explained.

Line 196 /  Figure 4

The arithmetic mean is not appropriate since Box-Whisker-Plots (and K-W-test) are used with not-normal distributed data. Median is appropriate.
Yes, we know. However, it may help from the different viewpoint; so that we want to add it in the box plots.

I can not follow this rational. Either values of a sample are normal distributed or not. The Box-Whisker-Plot indicates the distribution of measured values including outliers.

Line 211: Expelling of meconium has been described in the literature Smith, 1970, New & Theischinger 1993

Line 227: I know few Neuroptera species expelling fluid meconium (Mantispidae). Most families expel a solid pellet.

Line 254 ff: This chapter is speculative and allusive. There is no indication for lizards as predators in the literature.

Line 278: this trade-off is speculative since the data presented do not support this idea. However this hypothesis could be tested.

Line 296:  .. tends to decrease ..  data presented indicate that the decrease is statistically significant.

Author Response

Dear reviewer, we truly appreciate for your timely and careful revision.

All points suggested by you have been revised and the main changes are as follows:

Most of purely chemical defense have been deleted to simplify state expelling of meconial fluid in the Introduction and Discussion sections.

We have overlooked the very brief, but very important description in Smith (1970) “The fishfly Neohermes californicus (Walker) voids white meconium by just emerged adults and also field-captured adults when handled.” Therefore, we have revised the Introduction section to expand this behavior across Megaloptera. Thank you so much for your comment in this respect.

The number of individuals examined in Table 1 has been corrected, because the additional 7 individuals of Neochauliodes occidentalis in the last year were forgotten to add. Moreover, there were some individuals without the data of head width, which has been commented in the text as follows:

“The amount of expelled fluid varied among species and genera, and also among conspecific individuals (Table 1). The fresh weight of fluid (y mg) expelled by individuals was positively correlated with their head width (x mm) in the log-log relationship (log10 y = 3.159 log10 x – 1.614, r = 0.514, df = 277, p < 0.001) in all combined data (n = 279, excluding 2 Parachauliodes japonicus, 3 P. continentalis, 1 Neochauliodes occidentalis, 1 N. amamioshimanus, 1 Protohermes davidi, 2 P. grandis, and 1 Neoneuromus maclachlani lacking the data of head width among 290 expelled individuals in Table 1).”

Round 2

Reviewer 2 Report (Previous Reviewer 2)

general comments:

The manuscript has been improoved and can be accepted with minor revisions. The paragraph on statistics does not allow reproduction of calculations because the database is not indicated. Regarding Table 1 did not change false values that were noticed in the previous review.

To further improve the manuscript I added special comments that should be respected. Otherwise authors have to explain why they do not accept my comments.

special comments:

Line 33: delete „towards predators“

Line 45:  change  …. of meconial fluid might be one of such….

Line 70: divide in two sentences: …behavior is examined.  The function of
meconial expelling as defence against predators is discussed.

Line 73 (caption of Figure 1) add a, b, c ... to pictures and corresponding name in caption

Line 120: add  … once for individual adults bred in the lab.        

Line 161 ff (Table 1): The authors did not respect my previous comments of SE if n=2 altough they promised to have done. I really feel to be send up.

Line 176: avoid citation in Results chapter. This should be part of the Material and methods.

Line 179:  the amounts are not significantly different if p= 0.068

Line 183:  the amounts are not significantly different if p= 0.05 but when p<0.05

Line 227 ff: The difference between the behaviour of Megaloptera and defensive spraying by other insects has to be pointed out. For example: In contrast to the secretion from the anus by Mecoptera, several insects discharge fluids from specialized glands when disturbed. This has been shown for earwigs  ……

Line 233: delete first sentence (The chemical components of the defensive fluid may be species-specific [4].) There is no logical context to the text in this paragraph.

Line 247: change “produced” to “released”

Line 249: change:  … seems mostly to occur once

Line 257:  … fluid seems to affect predators ….

Line 259:  delete “important”

Line 260: delete the sentence: The fluid sprayed from the thoracic glands of a stick insect causes cleaning behaviour of lizards if it is attached to their head [28]. There is no logical context to the text in this paragraph.

Line 267:   change: The grasping of the thoracic region in our experiments may be similar to the stimulus by predators of Megaloptera.

Line 274 ff: This is a contradiction to the statement that the amount of expelled fluid correlates with body size. I suggest to delete line 274 to 279.

Line 279:  my suggestion to change the sentence: We need further information on morphology, physiology and behaviour in genera and species of Megaloptera to understand if there are avoidance mechanisms against predators.

Line 295: delete “as known in other insect groups that spray fluid against predators

Author Response

Dear Reviewer,

We are truly grateful for your timely and useful comments.

About our reply to the comments, please check the file.

With our best wishes to you.

Sincerely,

All authors

This manuscript is a resubmission of an earlier submission. The following is a list of the peer review reports and author responses from that submission.

Round 1

Reviewer 1 Report

Congratulations on a very original and interesting paper.  I am concerned whether "spraying" is the appropriate term, as the liquid or fluid appears to be rather as larger particles or amounts, not as small droplets such as in a spray or spraying event.  It would be interesting to do tests of taste in potential predators, as well as comparisons of amounts of expelling between long and short mandible taxa or representatives within a varying mandible length species.

Reviewer 2 Report

Comments and Suggestions for Authors

General comments

The authors describe two interesting phenomena in a predator-prey model: the release of meconial fluid by adult Corydalidae, the potential prey, and the deterrence of lizzards as predators. Both phenomena are described and experimental data (measurements) are presented. However, it is unclear what type of paper they intend to present: there is neither a testable hypothesis, controls are missing nor is there a discussion of a general research question. Finally, the authors link well documented phenomena to a theoretical predator-prey model and address the release of meconial fluid by Corydalidae as defensive spraying against predators.

The Article suffers from the misleading use of the term (defensive) spraying instead of release of meconial fluid or defecation as well as from the rational of the predator-prey-system. In the relevant literature (e.g. Eisner), (defensive) spraying is used to describe the release of any liquid from specialized glands aiming to defend the spraying item against antagonists. Spraying as chemical defence is contrasting defence by regurditation, easy blooding or release of faeces or even meconial fluid. The latter might also happen with some pressure to move the faeces from the individual. In the manuscript the different kinds of (chemical) defence by emitting substances are summarized as spraying and defecation and spraying from glands is mixed up in the discussion.

The authors use the predator-prey-system Mecoptera-lizzard as a model but do not give any evidence for its relevance. The literature mentions mainly birds or bats as Mecoptera-predators.

On the other hand, authors present well designed experiments both with Megaloptera and lizzards. The data of release and amount of meconial fluid by Mecoptera are novelties and should be published. The possible function as a deterrent might be discussed as a hypothesis.

In conclusion I reject the manuscript in the current form but suggest to write a paper on the release of meconial fluid of Corydalidae. Therefore, I added specific comments on this part only.

Specific comments

Line 58: Release of meconial fluid by Megaloptera was also mentioned by New & Theischinger 1993

Line 90: Would be nice to know if there are any observations of feeding on sucrose solution by females or males before release.

Line 92: Figure 1 does not show what is mentioned in the text but shows different species and the released meconial fluid.

Table 1: There are several critical points :

It is hardly possible to calculate SE from 2 values (even statistical programs produce values)

P. japonicas – 3 males + 10 females but 15/15 sprayed/examined individuals.

P continentalis – 32 males + 46 females but 69/83 sprayed/examined individuals

N. amamioshimanus – 16 + 12 but 21/29 sprayed/examined individuals

P. davidi – 10 + 8 but 17/19 sprayed/examined individuals

P. grandis – 15 + 24 but 35 / 43 sprayed/examined individuals

N. maclachlani –  2 + 5 but 8 / 8 sprayed/examined individuals

Would be nice to know the amount of meconial fluid of females and males. This data can be read from figure 3

Line 185: No reference for SPSS

Figure 4: The position of number of individuals should be in the same position in all diagrams

K-W-test is not appropriate, values are not independent since one specimen was sampled 3 times.

The arithmetic mean is not appropriate since Box-Whisker-Plots (and K-W-test) are used with not-normal distributed data. Median is appropriate.

Line 224 to 247 no comments on the experiments with lizzards

Discussion: see my general comments

Line 253:  ‘Spraying, not gentle secretion, occurs several body parts of insects.’ Meaning of the sentence

Line 266: However, the liquid sprayed by fishflies and dobsonflies is similar to the meconium sprayed from the anus by the noctuid moth Euxoa auxiliaris (Grote) when disturbed [17].  What is the rational of this statement (… similar to the meconium …)?  

Line 280: However, the contents other than meconium of the hindgut may be used because the sprayed liquid is occasionally transparent, whitish, or slightly reddish even in the same species. This is a very interesting point. Are there any observations about sex, age or feeding for individuals with ‘abnormal’ meconial fluid /faeces ?

Line 283 ff: The peak in quantity of released fluid just after emergence is a strong indication of its meconial origion and data do not support the idea of replenishing. The small quantity after some days might be retarded meconial fluid in the Malpighian tubules. However, this is a speculation.